# Azobenzene-bridged ionizable amphiphilic Janus glycosides for light-controlled, single-component and organ-modulable pDNA delivery

Zhaoxin Wang[1,6], Gonzalo Rivero-Barbarroja [ID][2,6], Juan M. Benito[3], Stéphane Maisonneuve[1], Itziar Vélaz[4], Inmaculada Juárez-Gonzálvez[5], María J. Garrido[5], Conchita Tros de Ilarduya[5], Carmen Ortiz Mellet [ID][2] ✉, Juan Xie [ID][1] ✉ & José M. García Fernández [ID][3] ✉

Stimuli-responsive supramolecular systems enable spatiotemporal control of nucleic acid (NA) delivery. To achieve precise and programmable vectors, we designed azobenzene-bridged ionizable amphiphilic Janus glycosides (IAJGs) as single-component, light-responsive DNA carriers. These glucopyranose-based dimers undergo reversible *E/Z* photoisomerization while forming stable nanocomplexes with plasmid DNA (pDNA). Photoisomerization alters nanocomplex size, surface charge, and internal order, resulting in distinct transfection outcomes. In vitro, *O*- and *S*-glycoside derivatives displayed isomer-dependent activity across COS-7, HepG2, and RAW264.7 cells, with pronounced switching effects specially in macrophages. In vivo, systemic administration revealed organ-selective responses: *O*-glycosides shifted expression from liver to lung upon $E \rightarrow Z$ conversion, whereas *S*-glycosides favored spleen targeting. All formulations maintained high cell viability. These results highlight photoswitchable IAJGs as structurally defined vectors for adjustable control over NA delivery and organ tropism.

Stimuli-responsive supramolecular systems offer chemists powerful means to exert spatiotemporal control over complex molecular assemblies and hold great promise in applications from sensing to drug delivery[1–4]. Their appeal is particularly strong when designed to interface with biological molecules, enabling dynamic modulation of function[5,6]. Nucleic acids (NAs) appear as particularly interesting targets for such a purpose[7–10]. Therapeutic NAs are reshaping modern medicine, yet their clinical impact is constrained by the challenge of safe, tissue-selective delivery[11]. Overcoming this barrier demands vectors that merge molecular precision with programmable control, enabling spatiotemporal regulation of cargo release. Viral systems, while highly efficient, face persistent concerns over immunogenicity, genomic integration, and manufacturing complexity[12,13]. Non-viral carriers that form dynamic nanocomplexes with the NA cargo offer a promising alternative to circumvent many of these limitations[14–16]. Within this family, four-component lipid nanoparticles (LNPs), encompassing ionizable lipids,

phospholipids, cholesterol, and polyethylene glycol (PEG)-conjugated lipids, have become the leading platform[17,18] and were instrumental in the rapid deployment of mRNA COVID-19 vaccines[19,20]. Nonetheless, their performance remains limited by inefficient endosomal escape (≈2%) and poor tissue specificity[21–24]. Improved efficiencies and extrahepatic organ selectivities have been attained with tailored LNP formulations incorporating either a fifth lipid[25–32], zwitterionic amino lipids[33] or alternative ionizable lipids identified via new chemistries couple with high-throughput screening[34,35]. Beyond LNPs, a common strategy to introduce responsiveness involves covalently incorporating functional groups, such as a photochrome, a pH-sensitive group or a redox species, into otherwise passive cationic polymers[8,9,36–40]. This versatile approach can generate virtually any user-defined, stimulus-responsive vector. However, the intrinsic multicomponent nature and polydispersity of such systems complicate precise structure-activity correlations, posing a major obstacle to rational design.

[1]Photophysique et Photochimie Supramoléculaires et Macromoléculaires, Université Paris-Saclay, ENS Paris-Saclay, CNRS, Gif-sur-Yvette, France. [2]Department of Organic Chemistry, Faculty of Chemistry, University of Seville, Seville, Spain. [3]Instituto de Investigaciones Químicas (IIQ), CSIC – Universidad de Sevilla, Seville, Spain. [4]Department of Chemistry, School of Sciences, University of Navarra, Pamplona, Spain. [5]Department of Pharmaceutical Sciences, School of Pharmacy and Nutrition, University of Navarra, Pamplona, Spain. [6]These authors contributed equally: Zhaoxin Wang, Gonzalo Rivero-Barbarroja. ✉e-mail: mellet@us.es; joanne.xie@ens-paris-saclay.fr; jogarcia@iiq.csic.es

Molecularly defined, single-component vectors offer a compelling solution[41,42]. Among then, ionizable Janus amphiphiles (IJAs) featuring sequence-defined multi-head/multi-tail architectures that can be finely tuned through precision chemistries, hold a leading position. IJAs built on macrocyclic scaffolds (e.g., cyclodextrins[43–49], cyclotrehalans[50–52], calixarenes[53–55]) and ionizable amphiphilic Janus dendrimers (IAJDs)[56–60] emerged as potent NA carriers, demonstrating programmable control over nanocomplex topology, intracellular trafficking, and tissue selectivity. This requires to adjust the IJA architecture for each system in order to elicit specific self-assembly and NA co-assembly patterns, making the approach case-specific. Introducing stimuli-responsiveness directly into these systems provides an exciting alternative, offering dynamic control over NA-vector assemblies without modifying the vector or the NA cargo themselves. Light is a particularly attractive trigger due to its remote actuation, tunability, and precise spatiotemporal control[61]. However, to our knowledge, transfection-competent, photoreversible supramolecular nanocomplexes involving nucleic acids and structurally defined delivery vectors have not yet been demonstrated. Azobenzene-containing molecules offer a convincing platform for such systems[62–64], owing to their robustness, rapid and reversible E/Z isomerization, and light-induced polarity and geometric changes that can modulate interactions with biomolecular targets[65–69]. Notably, cationic derivatives of azobenzene electrostatically interact with negatively charged mono- or polynucleotides[70], a process that has been used to photocontrol adenosine 5'-triphosphate (ATP) aggregation[71], DNA intercalation[72,73], G-quadruplex formation[74,75], DNA compaction[76–79] or gene expression[80,81], for instance, yet not within structurally defined NA carriers.

Most azobenzene-based systems rely on UV light for photoactivation, entailing phototoxic risks and restricting activation to superficial depths due to strong absorption and scattering by biological tissue. While this is acceptable for in vitro studies, translation to in vivo applications requires either safe light delivery into the body[82,83] or the development of azobenzene derivatives activatable under biocompatible irradiation[83–86]. Near-infrared (NIR) light is particularly attractive because of its deeper tissue penetration and low cytotoxicity[87,88]. One promising approach combines azobenzenes with upconverting nanoparticles (UCNPs) that absorb NIR light and emit higher-energy photons to trigger photoisomerization[83,88–91]. Nevertheless, even material-assisted strategies remain limited to centimeter-scale depths.

As a simpler and systemically compatible alternative, azobenzene agents can be pre-activated by UV irradiation prior to administration. This strategy is adopted here to provide a proof of concept for azobenzene-based, single-component vectors enabling light-modulable gene delivery in vitro and in vivo. Specifically, we introduce azobenzene-bridged IJA dimers that incorporate light-responsive switching into a twin-type D-glucopyranose-based vector construct. To ensure biological compatibility, we intentionally avoid conventional azobenzene-based cationic surfactants, which are prone to aggregation and precipitation in physiological media[92]. Instead, our design builds upon the previously established α,α′-trehalose-based molecular vector archetype. We have shown that the $C_2$ symmetry of this carbohydrate, formally a twin-glucodisaccharide, combined with the differential reactivity of its primary and secondary hydroxyl groups, enables the orthogonal installation of distinct ionizable headgroups and lipophilic tails with precise spatial orientation. This modularity permits systematic structural tuning to achieve organ-specific transfection in vivo. Building on this framework, we now demonstrate that twin ionizable amphiphilic Janus glycosides (IAJGs), consisting of azobenzene-bridged single-IAJ glucopyranosides, (i) undergo reversible E/Z photoisomerization ex vivo, (ii) form stable nanocomplexes with pDNA in both E and Z major states, and (iii) exhibit isomer-dependent transfection selectivity in vitro and organ-targeted delivery in vivo. Collectively, this work establishes a light-switchable, molecularly defined DNA delivery platform and defines optomechanical regulation as a strategy for programmable, stimulus-responsive nucleic acid therapeutics.

## Results and discussion
### Molecular design
We envisioned that twin-IAJGs **1** and **2** (Fig. 1a), comprising two identical glucose-based ionizable amphiphilic units bridged through a 4,4′-azobenzene module, would provide a robust platform for the formulation of light-responsive, single-component pDNA nanocomplexes. Direct glycosidic attachment of the sugar to the photochromic core was expected to

**Fig. 1 | Design of the photoswitchable twin-ionizable amphiphilic Janus glycosides (IAJGs). a** Chemical structures of the azobenzene O- and S-glucosides **1** and **2**. **b** Schematic illustration of the self-assembly of pDNA–IAJG nanocomplexes into lamellar, lipid bilayer-like architectures in the E state, and of the light-triggered nanomechanical isomerization process that induces local lipid phase transitions. An analogous nanocomplex can be formed from the Z-isomer, in which case transient destabilization of the supramolecular structure may similarly be achieved upon photo- or thermo-induced isomerization.

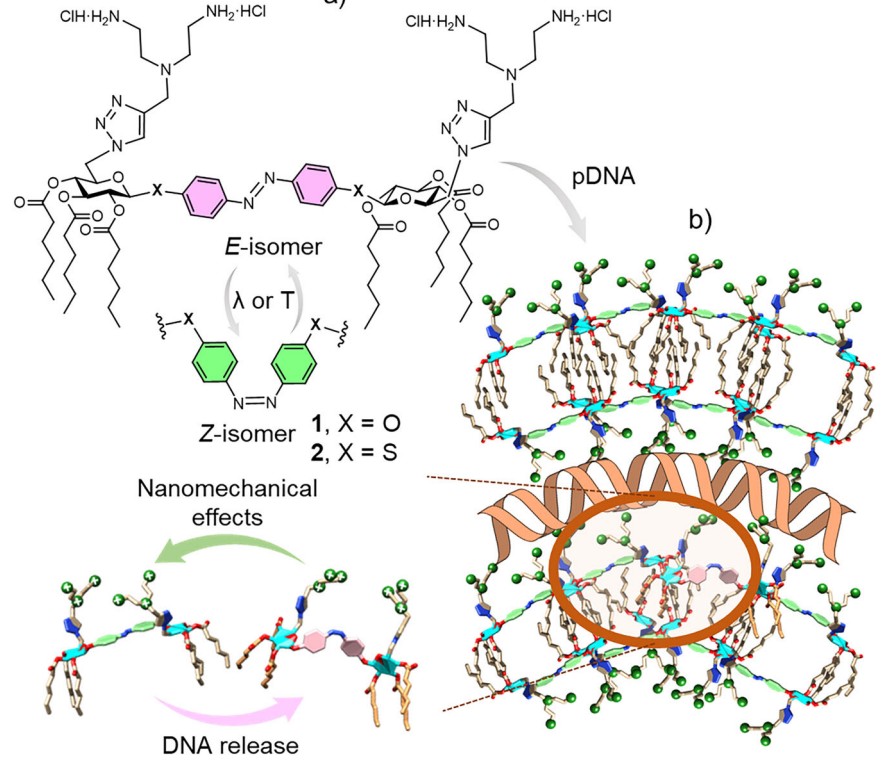

**Scheme 1 | Synthesis of IAJGs 1 and 2.** Reagents and conditions: **a** Tosyl chloride (TsCl; 1 eqv.), pyridine, 0 °C, 2 h, then r.t., overnight; **b** hexanoic anhydride (5 eqv.), pyridine, 0 °C→r.t., 1 h, 55–66% over two steps; **c** NaN$_3$, DMF, 70 °C, 16 h, 100%; **d** 9 (1.5 eqv.) CuSO$_4$·5H$_2$O (0.8 eqv.), sodium ascorbate (1.5 eqv.), CH$_2$Cl$_2$-H$_2$O = 3:1, r.t., 16 h, 65–68%; **e** CH$_2$Cl$_2$-TFA = 2:1, r.t., 1 h; **f** MeOH (0.1 mL), aq. HCl (0.1 M, 10 mL), freeze-drying, 100% over two steps.

maximize the transfer of conformational changes from the azobenzene hinge to the monosaccharide scaffolds. In parallel, the combination of triazole-linked dendritic diethylenetriamine headgroups, bearing both primary and tertiary amines, with hexanoyl tails has repeatedly proven effective in carbohydrate-based vectors, supporting strong pDNA condensation and efficient transfection both in vitro and in vivo[47,93,94]. The underlying mechanism reflects the dual contribution of electrostatic interactions with the polyanionic phosphate backbone and hydrophobic contacts between aliphatic domains of vector molecules, typically yielding nanocomplexes with long-range lamellar order characterized by alternating IAJG lipid bilayers and DNA segments. We anticipated that photoisomerization to the Z state would bring the two IAJG motifs of the dimer into closer proximity than in the E state, thereby altering self-assembly and co-assembly behavior. Such structural changes are expected to influence the internal order, topology, and surface charge of the resulting nanoparticles, parameters previously shown to govern differences in cell and organ transfection selectivity through passive targeting[45,51]. In addition, E ⇌ Z switching was anticipated to destabilize the nanocomplex ground state, independently of the initial isomer used in the formulation, by mechanically inducing phase transitions within the lipid bilayer domains[95–97], thereby facilitating endosomal escape through membrane disruption and promoting efficient nucleic acid release (Fig. 1b).

Transitioning from glycoside **1** to thioglycoside **2** was motivated by both functional and mechanistic considerations. Substitution of oxygen with sulfur increases resistance to enzymatic degradation by glycosidases in

biological media[98]. At the same time, differences in polarity, bond length, and electronic properties between O- and S-glycosides can influence the kinetics and thermodynamics of azobenzene isomerization, as well as the organizational stability of the resulting assemblies, broadening the opportunities for tuning the supramolecular properties, light responsiveness and cell/organ selectivities[99].

## Synthesis

To access the photoswitchable twin-IAJGs **1** and **2**, we implemented a divergent synthetic strategy starting from the 4,4′-bis(β-D-glucopyranosyl-oxy) and 4,4′-bis(β-D-glucopyranosyl-thio)azobenzene precursors **3** and **4**. Compound **3** was obtained via a protecting-group-free glycosylation of 4,4′-dihydroxyazobenzene with D-glucose, mediated by 2-chloro-1,3-dimethyl-limidazolinium chloride (DMC), as recently reported[100]. The thioglycoside precursor **4** was prepared through de-O-acetylation of the corresponding per-O-acetate, itself conveniently synthesized from 4,4′-diiodoazobenzene via a Buchwald–Hartwig–Migita cross-coupling with tetra-O-acetyl-1-thio-β-D-glucopyranose[101] (Scheme 1).

Subsequent sequential regioselective tosylation of the primary hydroxyls and hexanoylation of the secondary hydroxyls afforded intermediates **5** and **6**, which upon treatment with sodium azide yielded the azido-functionalized derivatives **7** and **8**. A copper(I)-catalyzed azide-alkyne cycloaddition (CuAAC) between **7** or **8** and 3-[N,N-di-[2-(N′-tert-butoxycarbonyl)aminoethyl]amino]prop-1-yne[94] (**9**) produced the corresponding Boc-protected triazole adducts **10** and **11**. Final deprotection of the

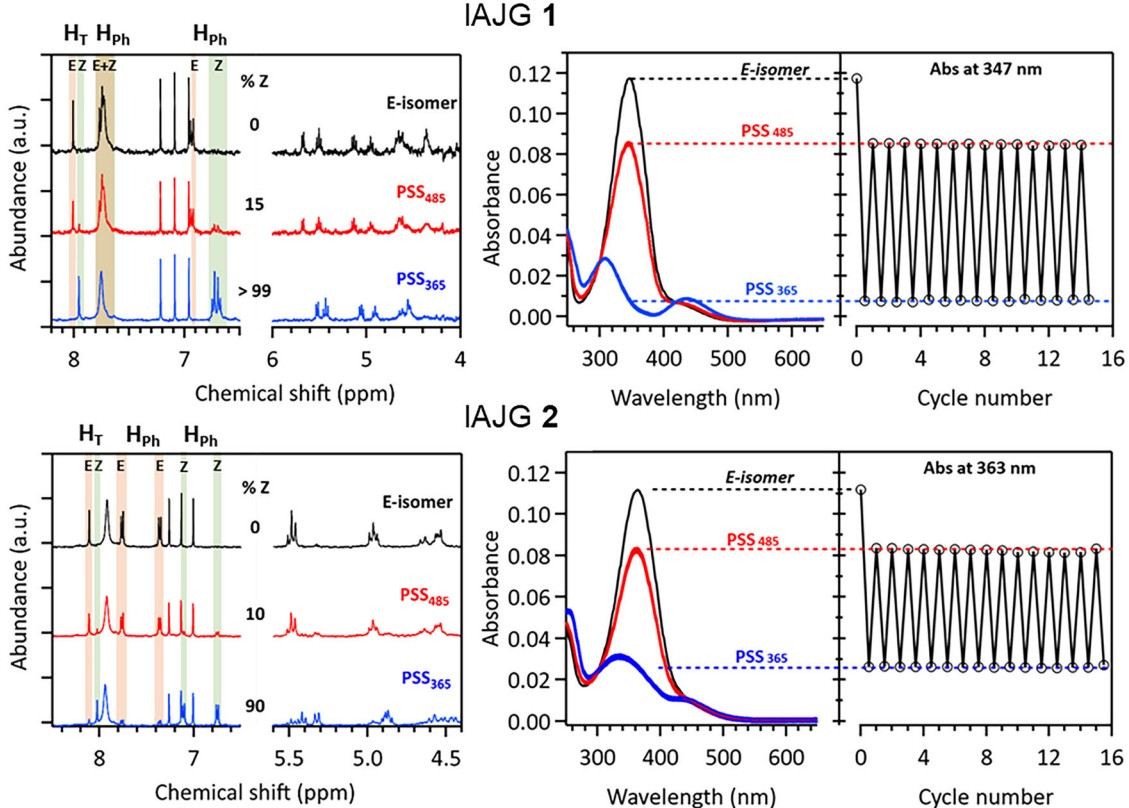

**Fig. 2 | Photochromic properties of IAJGs 1 (top panel) and 2 (bottom panel).** Left charts, aromatic and selected glucopyranoside regions of interest in the $^1$H NMR, recorded in DMSO-$d_6$, of the $E$-isomer (black line), PSS$_{485}$ (major $E$-isomer; red line) and PSS$_{365}$ (major $Z$-isomer; blue line). Diagnostic signals corresponding to the $E$ and $Z$ isomers are indicated by salmon- and light green-shaded backgrounds, respectively. The concentration was in the range of 0.1–0.2 mg·mL$^{-1}$ (33–66 μM).

Right charts, fatigue resistance monitored by UV–vis absorption in DMSO-$d_6$, showing the plot of the maximum absorption at 347 nm or 363 nm under alternate irradiation at 365 nm (IAJG **1**: 60 s, P = 7.5 mW·cm$^{-2}$; IAJG **2**: 30 s, P = 7.5 mW·cm$^{-2}$) and 485 nm (IAJG **1**: 300 s, P = 5.4 mW·cm$^{-2}$; IAJG **2**: 360 s, P = 5.4 mW·cm$^{-2}$; see also Supplementary Note 6, Fig. S20).

carbamate groups under acidic conditions delivered the target polyamines **1** and **2**, which were isolated as their tetrahydrochloride salts after lyophilization from dilute HCl (Scheme 1; full experimental details and characterization data, are provided in Supplementary Notes 1-3 and 11).

## Photoresponse of the azobenzene-bridged twin-IAJGs

With the two twin-IAJGs (compounds **1** and **2**) in hand, we next investigated their photochromic behavior in a 9:1 mixture of water and DMSO (see also Supplementary Notes 4–6, Figs. S18 and S19 and Tables S1 and S2). As illustrated in Fig. 2, both derivatives exhibit an initial absorption maximum ($\lambda_{max}$) at 347–363 nm, which originates from the π→π* transition of the predominant $E$ isomer. Upon irradiation at 365 nm, efficient $E$→$Z$ photoisomerization occurs, leading to a new absorption profile characterized by two main bands at 308–332 nm (π → π*) and 426–434 nm (n→π*). This corresponds to the photostationary state at 365 nm (PSS$_{365}$), in which the $Z$ form is dominant. Subsequent exposure to 485 nm light promoted the back isomerization ($Z$ → $E$), yielding PSS$_{485}$ that were enriched in the $E$ isomer. The back-switching process, however, proceeded with comparatively lower efficiency, a consequence of the spectral overlap between the two isomers in the 485 nm region. To quantify the $E/Z$ composition at each photostationary state, we performed $^1$H NMR spectroscopy in DMSO-$d_6$. The aromatic proton resonances of the $E$ and $Z$ isomers are well resolved, enabling reliable integration of their respective signals. Analysis of the irradiated samples confirmed the predominance of the $Z$ isomer at PSS$_{365}$ and of the $E$ isomer at PSS$_{485}$.

To evaluate the robustness of the switching process, we performed fifteen consecutive irradiation cycles, alternating between 365 nm (1 min)

and 485 nm (5 min). The absorbance maximum of the $E$ isomer at 347 nm (for **1**-$E$) or 363 (for **2**-$E$) was monitored after each cycle (Fig. 2). Importantly, no measurable loss of signal was detected, highlighting the excellent fatigue resistance of compounds **1** and **2**. Finally, the thermal back-isomerization ($Z$ → $E$) was followed by UV-Vis spectroscopy at 37 °C in a 9:1 mixture of water and DMSO (Fig. S19). 50% recovery of the $E$ isomer occurred after approximately 19.8 h for glycoside **1**-$Z$ and 30.6 h for thioglycoside **2**-$Z$, confirming their distinct thermal relaxation kinetics.

## Co-assembly of IAJGs with pDNA and IAJG/pDNA nanocomplex characterization

Having demonstrated selective access to both photoisomers of IAJGs **1** and **2**, we subsequently evaluated their capacity to induce pDNA nano-condensation and provide protection against degradation by nucleases. Particular attention was given to assessing whether the geometry of the azobenzene unit ($E$ vs $Z$) influences the co-assembly behavior, as well as the topology and stability of the resulting nanoparticles.

The ability of the tween-IAJG derivatives **1** and **2**, in both their $E$ and $Z$ major states, to form stable supramolecular nanocomplexes with pDNA (firefly luciferase-encoding pCMV-LucVR1216) was evaluated at protonable nitrogen-to-phosphorus (N/P) ratios of 5, 10, and 20 in HEPES buffer (10 mM, pH 7.4; see also the Supplementary Note 7 and Tables S3 and S4). DLS measurements (Table 1) confirmed the formation of self-assembled nanoparticles with positive ζ-potentials ranging from 12.9 to 33.9 mV. The average hydrodynamic diameter (D$_h$) systematically decreased with increasing N/P ratio, while the ζ-potential exhibited the opposite trend. Notably, the $Z$ isomers consistently yielded smaller nanocomplexes

(83–100 nm) than their *E* counterparts (100–135 nm) at the same N/P ratio. Differences in ζ-potential between the interconvertible *E* and *Z* states were particularly pronounced for the *O*-glycoside compound **1** (17–25 mV vs. 28–40 mV, respectively). In contrast, no clear correlation was observed for the corresponding *S*-glycoside **2**, where nanocomplexes formed with **2**-*Z* displayed significantly lower ζ-potential values than those obtained with **1**-*Z*. These results underscore the critical influence of vector structure, including both the glycosidic linkage nature and the azobenzene configuration, on nucleic acid nanocomplexation.

Formulations at N/P 10 were further examined by transmission electron microscopy (TEM) to gain insight into the morphology and internal organization of the nanocomplexes. The combined data confirm a strong correlation between the azobenzene geometry and the vector/pDNA co-assembly behavior. A direct comparison of the TEM micrographs obtained for **1**-*E* and **1**-*Z* clearly illustrates this effect. Nanocomplexes formed with **1**-*E* were larger and exhibited irregular spheroidal shapes, displaying only short-range order characterized by dark and white domains forming an intricate interlocking of arcs. By contrast, **1**-*Z*/pDNA complexes were smaller, more homogeneous, and showed well-defined ellipsoidal or spheroidal shapes with prominent long-range lamellar ordering of dark and light regions, likely corresponding to plasmid chains and lipid bilayer-type IAJG alignments, respectively. Even more striking differences were observed for the *S*-glycoside derivative: while **2**-*Z*/pDNA complexes closely resembled those obtained for **1**-*Z*, the corresponding **2**-*E* formulation gave rise to larger, globular structures, fully consistent with the DLS data (Fig. 3).

The ability of the different formulations to mediate pDNA complexation, protection, and preservation of overall pDNA integrity was subsequently assessed by electrophoresis mobility shift assay (EMSA) on 1% agarose gels stained with GelRed®. In all cases, successful pDNA binding and protection were evident, as demonstrated by the inhibition of plasmid migration within the gel, along with recovery of essentially intact pDNA following DNase I and sodium dodecyl sulfate (SDS) treatment (Supplementary Note 8, Fig. S21).

## Toxicity and in vitro cell transfection

Nanocomplexes formed from the twin IAJGs **1** or **2** and the pCMV-Luc VR1216 plasmid were assessed for their transfection performance in vitro using three representative cell lines: COS-7 (African green monkey kidney epithelial), HepG2 (human hepatocellular carcinoma), and RAW264.7 (murine macrophages). All transfections were performed in the presence of 10% serum. Briefly, the cells were incubated with the nanocomplexes for 4 h at 37 °C in the dark (internalization phase), then supplemented by fresh complete medium and further incubated for another 48 h, in the dark (protein expression phase). For comparison, polyplexes generated with branched polyethyleneimine (bPEI, 25 kDa), a benchmark cationic polymer widely employed in nonviral gene delivery, were included as controls. Absolute luciferase expression levels are shown in Fig. 4a. It should be noted that RAW264.7 cells are notoriously refractory to transfection, which complicates direct interpretation of efficiency across cell types and limits correlation between in vitro and in vivo outcomes[101]. Consequently, relative differences in activity are often more informative than absolute values[102]. To facilitate such comparisons, the results were normalized to the luciferase expression obtained with bPEI/pDNA polyplexes prepared at N/P 10, the optimal condition for the control in each cell line (Fig. 4b).

Transfection efficiency consistently increased as the N/P ratio rose from 5 to 10, an improvement attributable to the accompanying increase in particle surface charge, which likely strengthened electrostatic interactions with the negatively charged cell membrane. Substituting oxygen with sulfur at the glycosidic bonds produced cell-dependent effects on protein expression; for example, the *O*-glycoside derivative **1** outperformed the *S*-glycoside **2** in COS-7 (up to 3.6-fold higher for **1**-*Z* vs. **2**-*Z* at N/P 10) and HepG2 cells (up to 2.1-fold higher for **1**-*Z* vs. **2**-*Z* at N/P 10), whereas the opposite trend was observed in RAW264.7 macrophages (1.5-fold higher for **2**-*Z* vs. **2**-*E* at N/P 10). A parameter of specific interest is the shifting factor (SF), defined as the ratio of luciferase expression from nanocomplexes assembled with *Z*- vs *E*-isomers of a given IAJG, which exceeded unity in all experiments and reached values ranging from two to almost three orders of magnitude higher in RAW264.7 cells (Fig. 4c). Cytotoxicity, assessed by the AlamarBlue™ assay across all formulations and cell lines, showed that viability consistently remained above 80%, indicating no significant safety concerns in vitro (Supplementary Note 9 and Fig. S22).

The above data indicate that, beyond modifications to the IAJG vector backbone, the cell transfection profile can also be tuned by ex vivo photo-switching of the azobenzene module between its *E* and *Z* states, thereby introducing an additional layer of molecular diversity. An open question is whether the *Z*-isomer provides further benefit through an enhanced pDNA

**Table 1 | Average hydrodynamic diameter (D$_h$), polydispersity index (PDI), and ζ-potential values for nanocomplexes formulated with pCMV-LucVR1216 pDNA and IAJGs 1 and 2**

| Compound | N/P | D$_h$ (nm) | PDI | ζ-potential (mV) |
|---|---|---|---|---|
| **1**-*E* | 5 | 134.9 ± 1.1 | 0.212 ± 0.028 | +17.0 ± 0.9 |
| | 10 | 113.1 ± 10.9 | 0.113 ± 0.013 | +22.3 ± 0.7 |
| | 20 | 103.0 ± 1.1 | 0.154 ± 0.022 | +24.7 ± 0.6 |
| **1**-*Z* | 5 | 99.6 ± 5.9 | 0.151 ± 0.025 | +28.5 ± 0.7 |
| | 10 | 89.3 ± 2.0 | 0.113 ± 0.013 | +32.3 ± 3.0 |
| | 20 | 85.6 ± 3.8 | 0.161 ± 0.018 | +33.9 ± 1.8 |
| **2**-*E* | 5 | 124.9 ± 3.6 | 0.142 ± 0.006 | +20.8 ± 1.2 |
| | 10 | 149.1 ± 5.2 | 0.236 ± 0.049 | +21.9 ± 2.0 |
| | 20 | 100.3 ± 9.7 | 0.168 ± 0.012 | +22.6 ± 0.8 |
| **2**-*Z* | 5 | 99.0 ± 0.5 | 0.133 ± 0.064 | +12.9 ± 0.6 |
| | 10 | 89.1 ± 2.2 | 0.124 ± 0.034 | +19.7 ± 1.0 |
| | 20 | 83.6 ± 1.6 | 0.168 ± 0.006 | +25.6 ± 1.9 |

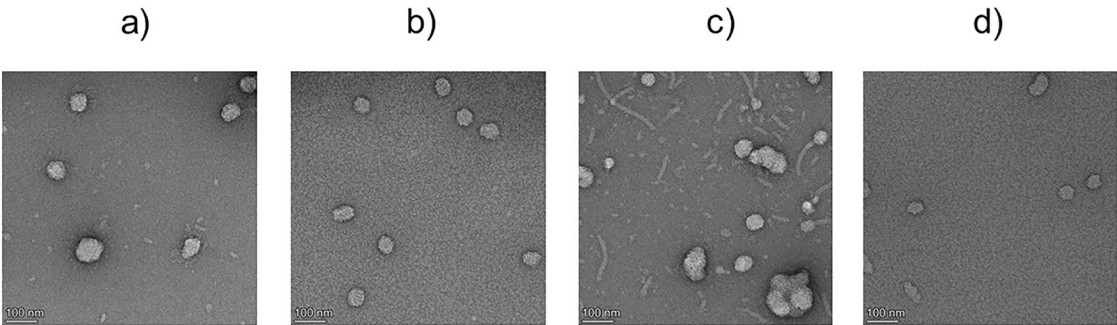

a) b) c) d)

**Fig. 3 | Representative TEM micrographs of N/P 10 IAJG/pDNA nanocomplexes. a** Formulations using **1**-*E*. **b** Formulations using **1**-*Z*. **c** Formulations using **2**-*E*. **d** Formulations using **2**-*Z*.

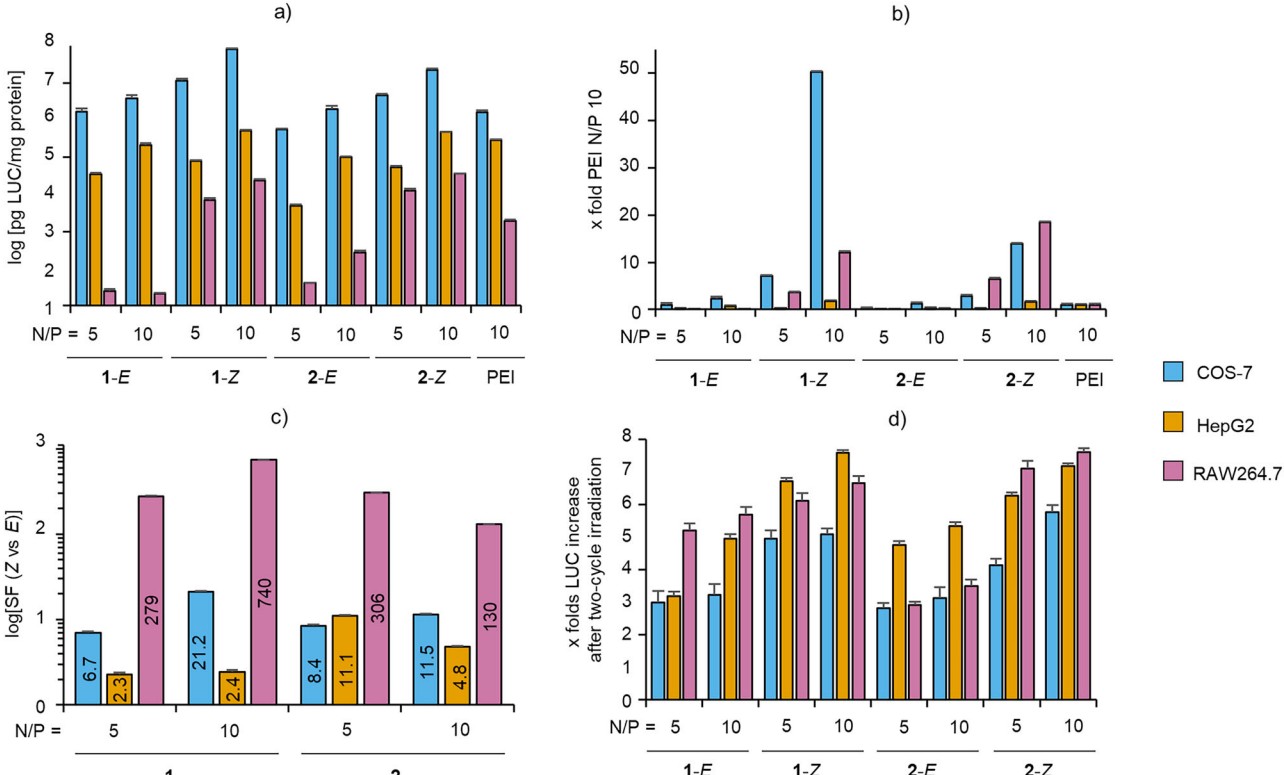

**Fig. 4 | In vitro transfection. a** Transfection efficiency in COS-7, HepG2 and RAW264.7 cells for nanocomplexes formulated with the IAJGs **1** and **2** in both the *E* or *Z* major form and the luciferase encoding plasmid pCMV-Luc VR1216 at N/P 5 and 10, in the presence of 10% fetal bovine serum (FBS). **b** Relative transfection efficiency data normalized to the value obtained for PEI/pDNA polyplexes (N/P 10) in each cell line. **c** Switching factor (SF) values, defined as the ratio between the transfection efficiency for the *Z* and the *E* isomer. **d** Relative transfection efficiency enhancement after a two alternating 5-min irradiation cycles at 365 nm/466 nm (for *E* isomers) or 466 nm/365 nm (for *Z* isomers). Blue, orange, and purple denote data obtained from COS-7, HepG2, and RAW264.7 cells, respectively. The data represent the mean ± SD (*n* = 3 biologically independent experiments).

release rate, driven by the thermally induced back-isomerization from *Z* to *E*. To ensure that maximum conversion is achieved, supplementary transfection experiments with *Z*-formulations were performed in which a 15-min UV irradiation (365 nm) was applied between the internalization and expression phases. The results demonstrated a modest yet consistent increase in luciferase expression across all three tested cell lines, with enhancements ranging from 1.3- to 2.3-fold. Strikingly, applying two alternating 5-min irradiation cycles at 365 nm/466 nm under identical conditions boosted transfection efficiency by 4.1- to 7.6-fold (Fig. 4d). Similarly, parallel experiments using *E*-formulations combined with two 466 nm/365 nm irradiation cycles yielded 2.8- to 5.7-fold improvements. Toxicity assays confirmed that these irradiation protocols did not compromise cell viability (Supplementary Note 9). In contrast, control PEI polyplexes showed no detectable effect on transfection. The ensemble of results discards a unidirectional mechanism, implying a more favorable pDNA release rate upon *Z* → *E* isomerization. Instead, they strongly suggest that isomer interconversion of the azobenzene module in the IAJGs, whatever direction causes permeability changes leading to enhanced cargo release. In agreement with data reported for azobenzene lipid-based liposomes[95–97], the nanomechanical action created by the isomerization further can promote the endosomal escape of the nanocomplexes, altogether leading to higher expressions of the encoded protein. Consistently, combined UV–Vis/DLS measurements after each irradiation step confirmed that isomerization occurs within intact nanocomplexes, accompanied by an increase in hydrodynamic size and a marked decrease in ζ-potential after completion of the four alternating irradiation steps, without appreciable disassembly. In cell culture medium prior to irradiation, all formulations exhibited a systematic increase in hydrodynamic size to

approximately 150 nm for the *E* isomers and 130 nm for the *Z* isomers, together with a decrease in surface charge. These changes are consistent with the formation of new species arising from interactions between the positively charged nanoplexes and negatively charged serum proteins[103]. Upon alternating irradiation, a moderate swelling, an increase in polydispersity index, and a further decrease in surface charge were observed, without evidence of precipitation or aggregation, supporting the proposed mechanism (Supplementary Note 12 and Tables S5 and S6). Importantly, the starting isomer dictates relative transfection efficiencies, even after the two-cycle irradiation cycle, underscoring nanoparticle topology as a key determinant of uptake and trafficking.

**In vivo cell transfection**

In vivo screening was carried out in mice following intravenous (50 µg/mouse, N/P = 10) administration of firefly luciferase pDNA via the tail vein. After 24 h, animals were sacrificed and luminescence quantified in homogenized organs (liver, heart, kidneys, lungs, spleen). Strong signals were detected in lung and liver for both **1**-*E*/pDNA and **1**-*Z*/pDNA nanocomplexes, with minimal expression in other tissues. Consistent with in vitro data, the *Z*-isomer increased overall protein output. Moreover, *E* → *Z* photoconversion further redirected organ selectivity from a liver-dominated profile (44%) to a lung-dominated one (61%). For the thioglycoside IAJG **2**, the *Z*-isomer again increased total transfection. Markedly different from **1**, however, organ tropism shifted from predominant lung expression (57%) to pronounced spleen selectivity (56%; Fig. 5a). The potential of pre-photoswitching of azobenzene-bridged tween-IAJGs, combined with molecular tailoring, to modulate passive organ targeting after systemic delivery is best captured by the corresponding *Z*/*E* SF values. For compound

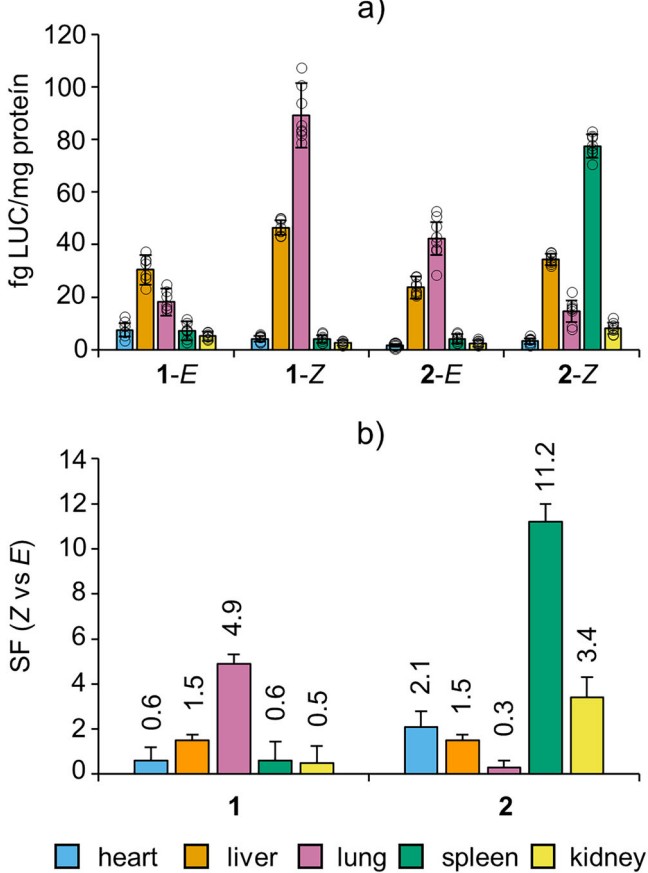

**Fig. 5 | In vivo transfection. a** Luciferase expression in the heart, liver, lungs, spleen and kidney of mice after intravenous administration of 50 μg of pCMV-Luc VR1216 formulated with **1** or **2**, in either the *E* or the *Z* major form. Black circles represent individual datapoints. **b** Switching factor (SF) values, defined as the ratio between the luciferase expression for the *Z* and the *E* major isomer in each organ. Blue, orange, purple, green and yellow denote data obtained from COS-7, HepG2, and RAW264.7 cells, respectively. Bars represent the mean ± SD (*n* = 8 animals).

**1**, *E* → *Z* isomerization boosts lung transfection nearly five-fold, whereas in compound **2** the same switch reduces efficiency to about one-third. Conversely, luciferase expression in the spleen decreases by roughly half for **1**-*E* relative to **1**-*Z*, but increases by more than an order of magnitude when comparing **2**-*E* to **2**-*Z* (Fig. 5b).

The variations observed in relative transfection efficiencies across cell types in vitro, as well as in organ tropism in vivo, are plausibly linked to differences in nanocomplex surface properties (shape and charge), internal structural organization, and their modulation upon photoswitching. For example, the preference of **1**-*Z* for COS-7 fibroblasts, which parallels its enhanced activity in the lung, can be attributed to the pronounced increase in surface charge following *E* → *Z* photoisomerization, accompanied by a transition to a lamellar arrangement. In contrast, the remarkable enhancement of transfection in macrophages and the spleen observed for **2**-*E* → **2**-*Z* switching appears to correlate with a substantial reduction in nanocomplex size and a shift from globular nanoparticles with short-range internal order to ellipsoidal or spheroidal particles exhibiting long-range lamellar organization. The molecular design and co-assembly of IAJGs are also expected to impart distinct endosomolytic properties. In particular, *Z*-isomer formulations may benefit from transient nanomechanical destabilization of the supramolecular edifice during thermal *E* → *Z* relaxation, favoring lysosomal escape and cargo release, consistent with the higher luciferase expression detected and the above discussed in vitro data. Nevertheless, correlations between in vitro and in vivo outcomes should be interpreted cautiously, as variability in transfection efficiency may also

reflect intrinsic differences in endocytic pathways or endosomal escape mechanisms shaped by both cellular context and vector structure.

## Conclusions
In conclusion, we have designed and synthesized azobenzene-bridged ionizable amphiphilic Janus glycosides (tween-IAJGs) as single-component, photoswitchable carriers for plasmid DNA. The IAJGs contain either *O*- or *S*-glycosidic linkages flanking an azobenzene bridge, enabling reversible and thermally stable *E/Z* isomerization upon UV–visible irradiation (365 and 466/485 nm). Both isomeric states form nanocomplexes with DNA, as inferred from formulations prepared from PSSs where one or the other is largely predominant, yet their supramolecular organization, size, and surface charge differ markedly after photoisomerization. These structural variations translate into distinct biological responses: in vitro, the *O*- and *S*-glycoside derivatives display cell-dependent differences in transfection activity, with particularly strong switching effects observed in macrophages; in vivo, light-induced isomerization prior to administration modulates organ tropism, shifting expression from liver to lung for *O*-glycosides and favoring spleen targeting for *S*-glycosides. Importantly, all formulations maintained high cell viability, indicating low cytotoxicity. Looking forward, the modular design of IAJGs offers opportunities to tailor photoswitchable carriers for targeted and externally controlled gene delivery, potentially extending their use to other therapeutic nucleic acids and precision nanomedicine applications.

## Methods
### Materials
General solvents and reagents were used without further purification. The reactions carried out under anhydrous conditions are performed under argon in glassware previously dried in an oven. Dimethylformamide (DMF) and toluene are previously dried by using a solvent purificator MBRAUN SPS-800. Dichloromethane ($CH_2Cl_2$) was dried over 4 Å molecular sieves, methanol (MeOH) was dried over 3 Å molecular sieves.

### Thin-layer chromatography (TLC)
Reactions were monitored by TLC on Silica Gel 60F-254 plates with detection by UV (254 nm or 365 nm) or by spraying with 10% $H_2SO_4$ in ethanol and heating about 30 s at 400–600 °C.

### Preparative column chromatography
Column chromatography purifications were performed on a CombiFlash® Rf+ purifier using RediSep® RF or RF Gold normal phase silica columns (with UV detection at 254 and 350 nm for all azobenzene-derivatives).

### High-performance liquid chromatography (HPLC)
The purity of IAJG compounds **1** and **2** was further assessed by HPLC. Briefly, an appropriate amount of the lyophilized sample was weighed into a vial and dissolved in prefiltered HPLC-grade methanol to obtain a 500 μM solution. An aliquot (1 mL) of this solution was then filtered through a 1 μm syringe filter into an HPLC vial. Chromatographic analyses were performed on an UltiMate 3000 system under the following conditions: XSelect CSH Phenyl-Hexyl reverse-phase column (4.6 × 100 mm, 3.5 μm particle size); column temperature, 30 °C; flow rate, 1.0 mL/min. The mobile phase consisted of a linear gradient from 95:5 water (0.1% formic acid)/acetonitrile (0.1% formic acid) to 100% acetonitrile (0.1% formic acid) over 12.5 min. Detection was carried out using a photodiode array (PDA) detector at 220 and 254 nm. The injection volume was 10 μL for all samples (Supplementary Note 10 and Figs. S23–S26).

### Optical rotation
Optical rotation [α]$_D$ was measured using a JASCO P-2000 polarimeter at 589 nm (Na light source) where no absorption occurred for all compounds, by using a 1.8 mm of aperture (S). The reported value corresponds to the average value from the acquisition of 3 iterations of 5 s (D.I.T.). The blank (solvent) was previously measured and automatically subtracted for each

sample. A cylindrical glass cell of 100 mm path length has been used for solution measurements (model CG3-100; 5 × 100 mm). The specific rotation $[\alpha]_a$ is based on the equation $[\alpha]_D = (100 \times \alpha) / (l \times c)$ where $[\alpha]_D$ is in (deg × mL) / (g × dm), the concentration c is in $g \cdot mL^{-1}$, and the path length l in dm.

## NMR spectroscopy

NMR spectra were recorded by using the JEOL ECS-400 spectrometer, Bruker Av400 or Bruker Av500. Irradiation frequencies of $^1H$ and $^{13}C$ were, respectively, 399.78, 400.08, 500.19 MHz and 100.53, 100.61, 125.79 MHz. The chemical shift was reported in delta (δ) parts per million (ppm) by using as reference the residual solvent signal or tetramethylsilane (TMS). The coupling constants (J) are given in Hertz (Hz). The followed abbreviations used are: singlet (s), doublet (d), triplet (t) and multiplet (m). The assignment was achieved by the help of 1D sequence for $^1H$, $^{13}C$, DEPT, and 2D sequences such as gCosy and gHMQC. UDEFT sequence was used preferentially for a low concentrated sample or for molecules containing $^{13}C$ atoms with long relaxation delay (Supplementary Note 2 and Figs. S1–S9). $^{19}F$ NMR spectra (282.4 MHz) were additionally recorded for IAJGs **1** and **2** to verify the absence of residual TFA in the final compounds. No detectable $^{19}F$ signals were observed, confirming complete removal of TFA. As a control, the $^{19}F$ NMR spectrum of compound **1** recorded prior to lyophilization from dilute HCl displayed the expected $^{19}F$ signals for the E (major) and Z (minor) isomers (Supplementary Note 11 and Fig. S27).

## HRMS spectrometry

High-resolution mass spectra (HRMS) were obtained on a Q-TOF mass spectrometer by the SALSA platform from the Institute of Organic and Analytical Chemistry (ICOA) – UMR 7311 (University of Orléans – CNRS; Supplementary Note 3 and Figs. S10–S17).

## UV-Vis spectroscopy

Absorption spectra were recorded on a Cary-5000 spectrophotometer from Agilent Technologies. A Hellma® quartz cell of 10 mm path length has been used for solution measurements.

## Photochromic reactions

The photochromic reactions were performed in quartz cuvette from Starna Scientific Ltd (type 23/Q/10; path length 10 mm), and induced in situ by a continuous irradiation with Hg/Xe lamp (Hamamatsu, LC8-Lightningcure, 200 W) equipped with narrow-band interference filters of 365 nm (Semrock FF01-365/XX-25) or 485 nm (Semrock FF01-485/20-25). The incident lamp power was measured by means of an Ophir PD300-UV photodiode. NIR contribution ($P_{LP}$) has been measured and subtracted from the total value using Schott long pass filter (LP-595) that is let through the Semrock filter ($P_{Total}$), and considering a 90% transmittance: $P_{\lambda irr} = P_{Total} - (10/9 \times P_{LP})$.

## Photoconversion yields

The photoconversion yields were measured from a solution of compound in deuterated solvent and monitored by $^1H$ NMR and UV-Vis absorption, after successive irradiations at 365 nm (485 nm) in the case of the PSS. The E:Z ratios were determined by integration of characteristic signals of each isomer. Data processing was realized with the help of Microsoft® Excel® and Igor Pro from WaveMetrics.

## Formulation of pDNA–IAJG nanocomplexes

The quantity of each IAJG vector used in each formulation was calculated based on the desired DNA concentration, the targeted protonable nitrogen-to-phosphorus (N/P) ratio, the molecular weight, and the number of protonable nitrogen atoms in the corresponding cationic derivative (Supplementary Note 7 and Tables S4 and S5). For E isomers, the stock solution was kept at 37 °C overnight. For Z isomers, the stock solution was irradiated at 365 nm for 2 h using a Hg/Xe lamp (Hamamatsu, LC8-Lightningcure) equipped with a narrow-band interference filter of the appropriate

wavelength (Hamamatsu A9616-07). The DNA solution phase was prepared by diluting a stock solution of pDNA in HEPES buffer (10 mM, pH 7.4, with 5% glucose) to the desired final concentration (0.005 µg/µL). The required amount of vector was then diluted from a stock solution in DMSO (typically 2 mM). The pDNA solution phase (900 µL) was transferred to an Eppendorf tube, and the vector solution phase (100 µL) was added. The resulting mixture (with a final DMSO content of 10% in all cases) was homogenized, and the complexes were incubated for 30 min under dark conditions prior to characterization in the case of E isomers, or measured immediately for Z isomers.

## Particle size and ζ-potential measurements

DLS/SLS of each pDNA/IAJG (**1** or **2**) formulation was measured in a Stunner nanoparticle analysis system (Unchained Labs) using 2 µL of samples in Stunner plate (96 wells). All Stunner measurements were performed with 4 replicates and 2 µL of sample. A buffer viscosity of 1.002 cP at 20 °C and 4 DLS acquisitions of 5 s each were used for DLS. Also, a Malvern Zetasizer Nano ZS instrument with back scattering detector (173°, 633 nm laser wavelength) was used for measuring the hydrodynamic size (diameter) in batch mode at 25 °C in a low volume quartz cuvette (path length 10 mm). Hydrodynamic size is reported as volume distribution of the major population by the mean diameter with its standard deviation. ζ-potential provides a measure of the electrostatic potential at the surface of the electrical double layer and the bulk medium, which is related to the nanoparticle surface charge. ζ-potentials measurements on the nanocomplexes were made using Malvern Zetasizer Nano ZS instrument with "mixed-mode measurement" phase analysis light scattering (M3-PALS). M3 consists of both slow field reversal and fast field reversal measurements, hence the name "mixed-mode measurement"; it improves accuracy and resolution. The following specifications were applied: sampling time, automatic; number of measurements, 12 per sample; medium viscosity, 1.002 cP; medium dielectric constant, 80; temperature, 25 °C.

## Electrophoresis mobility shift assay (EMSA)

Each vector:pDNA nanocomplex (20 µL, 0.1 µg of pDNA) was submitted to electrophoresis using a Gel Doc XR instrument (Bio-Rad, Hercules, CA, USA) for about 40 min under 80 V through a 1% agarose gel (with 10 µL of GelRed®) in TAE 1X buffer. The DNA was then visualized after photographing on a BioRad transilluminator. The fluorescence spectroscopy conditions were excitation at 482 nm and emission at 616 nm. The plasmid integrity in each sample was confirmed by electrophoresis after decomplexation with sodium dodecyl sulfate (SDS, 15%) and compared with that of the untreated DNA as a control.

## Transmission electron microscopy (TEM)

Samples were analyzed by electron microscopy after being adsorbed to glow-discharged Formvar-carbon coated grids and stained with 2% uranyl acetate. Grids were observed using a Thermo Fisher TALOS L120C transmission electron microscope operated at 120 kV. EM images were taken under low dose conditions with a Thermo Fisher CETA-F camera.

## In vitro transfection assay

Cells (COS-7, HepG2 or RAW264.7) were seeded in medium at a density of 6000 cells per well in 48-well plates (Iwaki Microplate, Japan) and incubated for 24 h at 37 °C in 5% $CO_2$. The culture plates for each cell line were divided into three groups (I–III). The medium was removed in the three groups, and 0.3 mL of complete medium (activated 10% FBS), 0.2 mL of nanocomplexes formulated with the IAJGs **1** or **2**, in either the E or the Z largely predominant form, or 25 kDa PEI polyplexes (N/P 10; control), containing 1 µg of pDNA. were added to each well and the plates were incubated for 4 h at 37 °C. Cells in group I were supplemented by fresh complete medium and further incubated for 48 h at 37 °C in the dark. Cells for group II were irradiated for 5 min under an UV lamp at 365 or 466 nm (for E- or Z-isomer formulations, respectively), then supplemented by fresh complete medium and further incubated for another 47 h and 45 min in the dark. Cells in

group III were subjected to four irradiation cycles of 5 min each using an UV lamp alternating 365/466 nm, then supplemented by fresh complete medium and further incubated for another 48 h in the dark. The power of the 365 and 466 nm are 5.4 and 20 mW·cm$^{-2}$, respectively. Cells in all groups were washed with phosphate-buffered saline (PBS) and lysed with 100 µL of Reporter Lysis Buffer (Promega, Madison, WI) at r.t. for 10 min, followed by a freeze-thaw cycle. A 20 µL aliquot of the supernatant was assayed for total luciferase activity by using the luciferase assay reagent (Promega), according to the manufacturer's protocol. A luminometer (Sirius-2, Berthold Detection Systems, Innogenetics, Diagnóstica y Terapéutica, Barcelona, Spain) was used to measure luciferase activity. The protein content of the lysates was measured by de DC protein Assay Reagent (Bio-Rad, Hercules, CA) with bovine serum albumin as the standard. The data were expressed as picograms of luciferase (based on a standard curve for luciferase activity) per milligram of protein. Samples were analyzed in a plate spectrophotometer Power Wave XS and a data processor KC junior, BioTek®.

### In vivo transfection assay
Balb-c mice (6–8 weeks of age, 20–25 g weight) were purchased from Harlan Ibérica Laboratories. Individual mice in groups of eight (four males and four females) were injected *via* the tail vein with 200 µL of nanoplexes containing 50 µg of pCMV-Luc VR1216 plasmid DNA at N/P 10. Naked DNA was injected as control. Twenty-four hours after injection, the mice were sacrificed. The liver, heart, kidneys, lungs, and spleen were collected and washed with cold PBS. We homogenized the organs with 1 mL of lysis buffer using a homogenizer at 5000 rpm (Mini-Beadbeater; BioSpec Products, Inc., Bartlesville, OK) and centrifuged at 10,000 rpm for 3 min. A 20 µL aliquot of the supernatant was analyzed for luciferase activity following the same procedure as for in vitro assays.

### Statistical analysis
Statistical analyses were performed using SPSS software from SPSS Inc. (Chicago, IL). The analysis of the transfection efficiency of the nanocomplexes was performed with a two-tailed unpaired Student's $t$-test. $P < 0.05$ was considered statistically significant.

### Ethical approval
This study was conducted following established ethical guidelines, in accordance with European (Directive 86/609/EEC) and national directives for protection of experimental animals, and approved by the Committee on Animal Research at the University of Navarra-CIMA (id ES/31-2010-000132; accreditation numbers CEEA 017-19 and 026-24). Our research team reflects a commitment to diversity and inclusion, with efforts made to support equal opportunities in scientific collaboration and authorship.

### Reporting summary
Further information on research design is available in the Nature Portfolio Reporting Summary linked to this article.

### Data availability
The authors declare that the data supporting the findings of this study related to chemical synthesis, photoswitching studies, and nanocomplex formulation are available within the paper and its Supplementary Information files. The raw data that support the biological findings of this study are available in Zenodo with the identifier https://doi.org/10.5281/zenodo.18223163.

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

## Acknowledgements

Z.W. gratefully acknowledges China Scholarship Council (CSC) for a doctoral scholarship. G.R.-B. acknowledges funding from the Ministerio de Ciencia, Innovación y Universidades through an FPU fellowship (Grant FPU18/02922). Additional support from the ENS Paris-Saclay Booster Program for mobility is also acknowledged. J.M.G.F. thanks Red de Enfermedades Raras CSIC (RER-CSIC) and PTI+ Salud Global – CSIC for continuous support. C.O.M. also acknowledge the CITIUS (Univ. Seville), for infrastructural support. The authors would like to thank Dr. Cyril Colas from the "Fédération de Recherche" ICOA/CBM (FR2708) for HRMS analysis. This work was supported by the Ministerio de Ciencia, Innovación y Universidades and the Agencia Estatal de Investigación, AEI/10.130 39/

501100011033 and "ERDF A way of making Europe" (PID2021-124247OB-C21, PID2021-124247OB-C22, PID2022-141034OB-C21, PID2024-157753OB-C21 and PID2024-157753OB-C22). We also acknowledge funding by the European Union's Horizon Europe research and innovation program under the Marie Skłodowska-Curie grant agreement 101130235 – Bicyclos.

## Author contributions

Z.W. carried most of the synthesis and the photocharacterization experiments; G.R.-B. and J.M.B. contributed to the synthesis and photocharacterization and also conducted the formulation of the nanococomplexes and their physicochemical characterization; S.M. contributed to curation and analysis of the photochemical data; I.V., I.J.-G., M.J.G., and C.T.del. performed and analyzed the biological assays; all authors contributed to experimental design, data analysis and manuscript edition; J.M.B., S.M., C.T.del., C.O.M., J.X., and J.M.G.F. wrote the paper. C.O.M., J.X., and J.M.G.F. devised and supervised the work.

## Competing interests

The authors declare no competing interests.
