## [Transparent Peer Review file · Communications Chemistry]

Azobenzene-Bridged Ionizable Amphiphilic Janus Glycosides for Light-Controlled, Single-Component and Organ-Modulable pDNA Delivery

Corresponding Author: Professor Jose Garcia Fernandez

Version 0:

Reviewer comments:

Reviewer #1

(Remarks to the Author)

The manuscript describes a chemical approach to the preparation of gene-delivery systems which can influence DNA binding and release upon UV light irradiation. Although the approach is not completely novel, the results collected clearly indicate that the DNA binding, compactation and delivery is influenced by the E/Z configuration of the azobenzene included in the vector. The work is well described and the statistical analysis properly carried out. Compounds are properly characterized and experimental procedures properly described. Data collected are solid and justify the conclusions given by the authors. Due to the general interest in obtaining novel responsive gene-delivery systems and to the rigorous chemical approach used by the authors, I think that the manuscript is certainly of interest to the chemical community and worth publishing in Communications Chemistry after the following revisions.

In the introduction, authors should comment on the use of such systems in vivo and in particular on the feasibility of Z/E isomerization by UV irradiation in a living system/organ.

The authors interestingly highlighted how the aggregate morphologies with pDNA differ when compounds 1 and 2 are pre-isomerized mainly into either the E or Z form before contacting DNA. It would also be worthwhile to investigate what happens to the morphology of the E-form aggregates with pDNA when they are irradiated to induce isomerization to the Z form. Do the aggregates change their morphology, suggesting that isomerization can also occur when compounds 1 and 2 are already bound to DNA or the interaction of the vectors with DNA is so strong that isomerization is prevented once the vector is bound?

The term *ex vivo* is used along the manuscript (three cases) but I think the term *in vivo* is a more appropriate terminology to describe those experiments.

In fig. 2 and its caption, a PSS370 is indicated while in the text is always used the term PSS365. Is there a difference?

Line 70: authors should cite few papers where azobenzene is used to modulate gene-delivery. At least, authors should quote 10.1016/j.colsurfb.2016.07.028 and 10.1016/j.colsurfb.2016.07.028

References 9 and 38 are duplicated, so as for 8 and 42.

Reviewer #2

(Remarks to the Author)

In this manuscript, Z. Wang et al. have synthesized two azobenzene-bridged ionizable amphiphilic Janus glycosides, characterized their photochromic properties, and evaluated them as light-responsive DNA carriers. Importantly, the authors performed studies on three different cell lines treated with Z-rich or E-rich mixtures, and also carried out in situ photoswitching. Notably, the nanoparticle behavior observed in vitro was further confirmed in mouse studies. I recommend this manuscript for publication after the authors address the following issues.

1. Please provide the ¹⁹F NMR spectra for the two final compounds to confirm that all TFA has been removed. Since these compounds are isolated as salts, elemental analysis should also be provided. In addition, please perform HPLC analysis for final compounds with detection at 220 nm and 254 nm.

2. Because the authors irradiated the cells with 365 nm light ("To ensure that maximum conversion is achieved,

supplementary transfection experiments with Z-formulations were performed in which a 15-min UV irradiation (365 nm) was applied between the internalization and expression phases.”), they should also perform a control experiment to evaluate the effect of 365 nm light on the cells, particularly to assess cell viability after light exposure.

3. The statement in the conclusion is incorrect (“enabling reversible and thermally stable E/Z isomerization upon irradiation with visible light”), because 365 nm is not visible light, and the authors use this wavelength to induce the E → Z conversion. Please revise this sentence accordingly.

4. Please add data to Table 1 (or the Supporting Information) showing the nanoparticle parameters after the switching cycles. Since the nanocomplexes underwent two alternating 5-min irradiation cycles at 365 nm and 485 nm, which affected transfection efficiency, it is important to determine whether these changes are due to the switching cycles themselves or simply a result of the cellular environment. If possible, please perform this experiment in cell culture media.

Minor edits:

5. In Figure 2, “PSS 365” should be used instead of “PSS 370.” Please correct this. Additionally, the current version of Figure 2 is difficult to read, the scale, particularly in the aliphatic region of the NMR spectrum (bottom), is unclear. Please revise the figure to improve legibility. Furthermore, indicate explicitly which signals were compared (e.g., label them with “Z” and “E,” or use another clear notation) so that readers can easily understand which peaks were used to calculate the ratio. Please also include the sample concentration in the NMR description.

6. Should the wavelength be 485 nm instead of 456 nm? This appears to be a recurring mistake throughout the manuscript (e.g., “Similarly, parallel experiments using E-formulations combined with two 456 nm/365 nm...”). Please verify and correct all instances.

7. Please correct the English in the Supporting Information, as I found several errors (e.g., “acquisition,” “an extemporaneous solution,” “the PSS is validated,” etc.).

Version 1:

Reviewer comments:

Reviewer #1

(Remarks to the Author)

The authors have satisfactorily addressed all reviewers’ questions and criticisms, making the appropriate revisions to both the manuscript and the Supporting Information. The manuscript now meets the required standards of clarity and scholarly presentation and is suitable for publication without further delay.

Reviewer #2

(Remarks to the Author)

The authors have now addressed all of my comments and concerns. I have no further remarks. In its present form, the manuscript is suitable for publication in Communications Chemistry.

Answers to reviewers' comments

Reviewers' comments:

Reviewer #1 (Remarks to the Author):

The manuscript describes a chemical approach to the preparation of gene-delivery systems which can influence DNA binding and release upon UV light irradiation. Although the approach is not completely novel, the results collected clearly indicate that the DNA binding, compactation and delivery is influenced by the E/Z configuration of the azobenzene included in the vector. The work is well described and the statistical analysis properly carried out. Compounds are properly characterized and experimental procedures properly described. Data collected are solid and justify the conclusions given by the authors. Due to the general interest in obtaining novel responsive gene-delivery systems and to the rigorous chemical approach used by the authors, I think that the manuscript is certainly of interest to the chemical community and worth publishing in Communications Chemistry after the following revisions.

1.- In the introduction, authors should comment on the use of such systems *in vivo* and in particular on the feasibility of Z/E isomerization by UV irradiation in a living system/organ.

Answer: We appreciate the reviewer's suggestion to address the use of azobenzene-based systems *in vivo*, particularly regarding the feasibility of Z/E isomerization by UV irradiation in living organisms. We agree that this is an important consideration. The main challenges in *in vivo* applications involve mitigating phototoxicity and enhancing tissue penetration. Current strategies focus on overcoming these limitations by utilizing near-infrared (NIR) irradiation, either through the modification of azobenzene structures or by incorporating upconverting nanoparticles to facilitate deeper tissue penetration. To address this, we have added a brief paragraph in the **Introduction** section that discusses these challenges, referencing key reviews and primary studies, just before summarizing the scope of our results:

"Most azobenzene-based systems rely on UV light for photoactivation, entailing phototoxic risks and restricting activation to superficial depths due to strong absorption and scattering by biological tissue. While this is acceptable for *in vitro* studies, translation to *in vivo* applications requires either safe light delivery into the body^{82,83} or the development of azobenzene derivatives activatable under biocompatible irradiation⁸³⁻⁸⁶. Near-infrared (NIR) light is particularly attractive because of its deeper tissue penetration and low cytotoxicity^{87,88}. One promising approach combines azobenzenes with upconverting nanoparticles (UCNPs) that absorb NIR light and emit higher-energy photons to trigger photoisomerization^{88,83,89-91}. Nevertheless, even material-assisted strategies remain limited to centimeter-scale depths.

As a simpler and systemically compatible alternative, azobenzene agents can be pre-activated by UV irradiation prior to administration. This strategy is adopted here to provide a proof of concept for azobenzene-based, single-component vectors enabling light-modulable gene delivery *in vitro* and *in vivo*".

2.- The authors interestingly highlighted how the aggregate morphologies with pDNA differ when compounds 1 and 2 are pre-isomerized mainly into either the E or Z form before contacting DNA. It would also be worthwhile to investigate what happens to the morphology of the E-form aggregates with pDNA when they are irradiated to induce isomerization to the Z form. Do the aggregates change their morphology, suggesting that isomerization can also

occur when compounds 1 and 2 are already bound to DNA or the interaction of the vectors with DNA is so strong that isomerization is prevented once the vector is bound?

Answer: We thank the reviewer for this insightful suggestion, which was also raised by Reviewer 2. To address whether photoisomerization can occur once compounds 1 and 2 are already complexed with pDNA—and whether this affects aggregate morphology—we performed two complementary sets of experiments:

(a) We simultaneously recorded DLS/SLS and UV–vis data of the nanocomplexes before and after each irradiation step using a Stunner instrument (Unchained Labs). These measurements demonstrate efficient and reversible *E/Z* isomerization at each irradiation cycle, accompanied by an increase in hydrodynamic size and a marked decrease in ζ -potential after completion of the four alternating irradiation steps.

(b) In parallel, we monitored the average hydrodynamic diameter and ζ -potential in DMEM (Dulbecco's Modified Eagle Medium) supplemented with 10% fetal calf serum (FCS) to mimic cell culture conditions. Measurements were performed using a Zetasizer Nano (Malvern) before irradiation and immediately after completion of the irradiation cycles. Compared with water, all formulations in cell culture medium showed a systematic increase in hydrodynamic size to approximately 150 and 130 nm for the *E* and *Z* isomers, respectively, along with a decrease in surface charge, with no evidence of precipitation or formation of larger aggregates. These changes are consistent with the formation of new species arising from interactions between the positively charged CDplexes and negatively charged serum proteins. Upon alternating irradiation, a moderate swelling of the nanocomplexes, accompanied by an increase in polydispersity index and a decrease in surface charge, was observed.

Taken together, these results indicate that azobenzene photoisomerization remains operative after DNA binding and does not cause nanocomplex disruption. Rather, the reversible physicochemical changes support a light-mediated, transient nanomechanical effect that may facilitate pDNA release while preserving overall aggregate integrity.

A paragraph has been incorporated at the end of the **Toxicity and *in vitro* cell transfection** subsection to summarize this information, and the corresponding data are collected in the Supplementary material:

“Consistently, combined UV–vis/DLS measurements after each irradiation step confirmed that isomerization occurs within intact nanocomplexes, accompanied by an increase in hydrodynamic size and a marked decrease in ζ -potential after completion of the four alternating irradiation steps, without appreciable disassembly. In cell culture medium prior to irradiation, all formulations exhibited a systematic increase in hydrodynamic size to approximately 150 nm for the *E* isomers and 130 nm for the *Z* isomers, together with a decrease in surface charge. These changes are consistent with the formation of new species arising from interactions between the positively charged nanoplexes and negatively charged serum proteins¹⁰³. Upon alternating irradiation, a moderate swelling, an increase in polydispersity index, and a further decrease in surface charge were observed, without evidence of precipitation or aggregation, supporting the proposed mechanism (Supplementary Note 12 and Tables S5 and S6)”.

3.- The term *ex vivo* is used along the manuscript (three cases) but I think the term *in vivo* is a more appropriate terminology to describe those experiments.

Answer: We acknowledge that the use of the term *ex vivo* in the manuscript may have caused confusion. In our study, this term was intended to indicate that UV irradiation of the pDNA nanocomplexes and photoswitching of the twin-IAJG molecular vector were performed prior to *in vivo* evaluation in mice, rather than in a biological system. Upon reconsideration, we have revised the manuscript as follows: (i) the first occurrence of *ex vivo* in the **Introduction**, where its use was unnecessary, has been removed; (ii) in the **Toxicity and in vitro cell transfection** subsection, the term was incorrectly used and has been replaced by “prior”; and (iii) in the **In vivo cell transfection** subsection, it has been replaced with “pre-photoswitching of azobenzene-bridged tween-IAJGs” to more precisely describe the experimental procedure. These changes clarify the terminology and avoid potential ambiguity.

4.- In fig. 2 and its caption, a PSS370 is indicated while in the text is always used the term PSS365. Is there a difference?

Answer: Thank you for pointing this out. This is indeed an error, also noted by Reviewer 2. Although the optical filter used has a bandwidth centered at 370 nm, the transmitted light is effectively centered at 365 nm. The correct designation is therefore PSS₃₆₅. Figure 2 has been corrected accordingly in the revised manuscript. See also the answer to comment 5 from Reviewer 2 for description of further improvements made on this Figure.

5. Line 70: authors should cite few papers where azobenzene is used to modulate gene-delivery. At least, authors should quote 10.1016/j.colsurfb.2016.07.028 and 10.1016/j.colsurfb.2016.07.028

Answer: We thank the reviewer for this helpful suggestion. In an earlier version of the manuscript, a dedicated paragraph and several references addressing azobenzene-based macromolecular and host–guest systems for gene delivery were included but were later removed for conciseness, as the main focus of the paper is on molecularly defined vectors. Nevertheless, we agree that citing representative examples in this context is appropriate. Accordingly, we have now added the references suggested by the reviewer at the indicated position in the text, together with two additional relevant studies, without requiring any rewording of the surrounding content (references 62-64 in the revised version).

6. References 9 and 38 are duplicated, so as for 8 and 42.

Answer: We apologize for this oversight. The duplicated references (9 and 38, as well as 8 and 42 in the original submission) have now been corrected in the revised manuscript.

Reviewer #2 (Remarks to the Author):

In this manuscript, Z. Wang et al. have synthesized two azobenzene-bridged ionizable amphiphilic Janus glycosides, characterized their photochromic properties, and evaluated them as light-responsive DNA carriers. Importantly, the authors performed studies on three different cell lines treated with Z-rich or E-rich mixtures, and also carried out *in situ* photoswitching. Notably, the nanoparticle behavior observed *in vitro* was further confirmed in mouse studies. I recommend this manuscript for publication after the authors address the following issues.

1. Please provide the ¹⁹F NMR spectra for the two final compounds to confirm that all TFA has been removed. Since these compounds are isolated as salts, elemental analysis should also be

provided. In addition, please perform HPLC analysis for final compounds with detection at 220 nm and 254 nm.

Answer: We thank the reviewer for raising this point. We routinely record ^{19}F NMR spectra for molecular vectors when TFA is used in the final synthetic step, as TFA counterions can form strong ion pairs with ammonium cations and significantly affect DNA condensation and transfection efficiency. We now provide the corresponding ^{19}F NMR spectra for compounds **1** and **2**, which show no detectable ^{19}F signals, confirming the absence of TFA. As a control, we include the ^{19}F NMR spectrum of compound **1** prior to lyophilization from dilute HCl, displaying the expected signals for the *E* (major) and *Z* (minor) isomers. The following text was incorporated at the end of the **NMR spectroscopy** subsection:

“ ^{19}F NMR spectra (282.4 MHz) were additionally recorded for IAJs **1** and **2** to verify the absence of residual TFA in the final compounds. No detectable ^{19}F signals were observed, confirming complete removal of TFA. As a control, the ^{19}F NMR spectrum of compound **1** recorded prior to lyophilization from dilute HCl displayed the expected ^{19}F signals for the *E* (major) and *Z* (minor) isomers”.

Copies of the ^{19}F NMR spectra are also incorporated in the Supplementary information, Note 11 and Figures S27.

In addition, combustion analysis of the same samples yielded microanalytical data for C, H, N (compound **1**) and C, H, N, S (compound **2**) in good agreement ($\pm 0.4\%$) with the expected tetrahydrochloride salts. The data have been incorporated in the Supplementary information, at the end of the corresponding spectroscopic and MS data:

For **1**: “Elemental analysis, Found: 55.12; H, 7.32; N, 11.97. Calc. for $\text{C}_{74}\text{H}_{122}\text{Cl}_4\text{N}_{14}\text{O}_{16}$: C, 55.35; H, 7.66; N, 12.21”.

For **2**: “Elemental analysis, Found: 53.95; H, 7.20; N, 11.64; S, 3.73. Calc. for $\text{C}_{74}\text{H}_{126}\text{N}_{14}\text{O}_{14}\text{S}_2$: C, 54.27; H, 7.51; N, 11.97; S, 3.92”.

Following the reviewer’s suggestion, we also performed HPLC analysis, which supports purities $>96\%$ for compound **1-E** and $>92\%$ for compound **2-E**, considering the main peak in the chromatogram. A dedicated subsection has been incorporated in the main manuscript, after the **Preparative column chromatography** subsection:

“High-Performance Liquid Chromatography (HPLC)

The purity of IAJG compounds **1** and **2** was further assessed by HPLC. Briefly, an appropriate amount of the lyophilized sample was weighed into a vial and dissolved in prefiltered HPLC-grade methanol to obtain a 500 μM solution. An aliquot (1 mL) of this solution was then filtered through a 1 μm syringe filter into an HPLC vial. Chromatographic analyses were performed on an UltiMate 3000 system under the following conditions: XSelect CSH Phenyl-Hexyl reverse-phase column (4.6 \times 100 mm, 3.5 μm particle size); column temperature, 30 $^\circ\text{C}$; flow rate, 1.0 mL/min. The mobile phase consisted of a linear gradient from 95:5 water (0.1% formic acid)/acetonitrile (0.1% formic acid) to 100% acetonitrile (0.1% formic acid) over 12.5 min. Detection was carried out using a photodiode array (PDA) detector at 220 and 254 nm. The injection volume was 10 μL for all samples (Supplementary Note 10 and Fig. S23-S26).”

Additionally, copies of the HPLC chromatograms with detection at 220 nm and 254 nm have been incorporated in the Supplementary information, Note 10 and Figures S23-S26.

2. Because the authors irradiated the cells with 365 nm light (“To ensure that maximum conversion is achieved, supplementary transfection experiments with Z-formulations were performed in which a 15-min UV irradiation (365 nm) was applied between the internalization and expression phases.”), they should also perform a control experiment to evaluate the effect of 365 nm light on the cells, particularly to assess cell viability after light exposure.

Answer: We thank the reviewer for this important comment. The requested control experiment was in fact already performed: cell viability was assessed after exposure to alternating 365 nm irradiation under the same conditions used in the transfection experiments, and no statistically significant changes in viability were observed. Please, note that there was a mistake in the description of the experimental procedure: irradiation times are 5 min (as indicated in the **Discussion**, not 15 min. While the absence of toxicity was mentioned in the **Conclusions**, we agree that this information should be explicitly reported in the **Results and Discussion** section. Accordingly, we have now added the following sentence after the discussion of the irradiation experiments:

“Toxicity assays confirmed that these irradiation protocols did not compromise cell viability (Supplementary Note 9)”.

Additionally, in the Supplementary information, Note 9 (Cell Viability Assay), the following sentence was added:

“Cell viability was similarly assessed after applying four irradiation cycles of 5 min each using a UV lamp with alternating 365/466 nm irradiation. No statistically significant differences in cell viability were observed compared with non-irradiated control experiments”.

3. The statement in the conclusion is incorrect (“enabling reversible and thermally stable E/Z isomerization upon irradiation with visible light”), because 365 nm is not visible light, and the authors use this wavelength to induce the E → Z conversion. Please revise this sentence accordingly.

Answer: You are correct, and we apologize for the mistake. Since the E→Z conversion is induced at 365 nm (UV), the statement “visible light” was inaccurate. We have revised the sentence to refer to UV–visible irradiation (365 and 485 nm) and corrected it accordingly in the revised manuscript.

4. Please add data to Table 1 (or the Supporting Information) showing the nanoparticle parameters after the switching cycles. Since the nanocomplexes underwent two alternating 5-min irradiation cycles at 365 nm and 485 nm, which affected transfection efficiency, it is important to determine whether these changes are due to the switching cycles themselves or simply a result of the cellular environment. If possible, please perform this experiment in cell culture media.

Answer: Please see our response to Reviewer 1 (comment 2) for a detailed description of the additional experiments performed to address this suggestion and the corresponding revisions to the manuscript. The requested experimental data are provided in Supplementary Note 12 and Tables S5 and S6.

Minor edits:

5. In Figure 2, "PSS 365" should be used instead of "PSS 370." Please correct this. Additionally, the current version of Figure 2 is difficult to read, the scale, particularly in the aliphatic region of the NMR spectrum (bottom), is unclear. Please revise the figure to improve legibility. Furthermore, indicate explicitly which signals were compared (e.g., label them with "Z" and "E," or use another clear notation) so that readers can easily understand which peaks were used to calculate the ratio. Please also include the sample concentration in the NMR description.

Answer: Thank you for pointing out the issues with this figure. We have revised it accordingly. Specifically, the following improvements were implemented: (a) correction of the photostationary state designation to PSS₃₆₅ (see also our response to Reviewer 1, Comment 4); (b) clear identification of the NMR signals corresponding to the *E* and *Z* isomers using different colors, with explicit labels "*E*" (orange) and "*Z*" (green); and (c) adjustment of the scale tick intervals to improve readability. Regarding the NMR experiments, the concentration used for the NMR experiment was in the range of 0.1-0.2 mg/mL. The following text was inserted in Figure 2 legend to specify this:

"The concentration was in the range of 0.1–0.2 mg·mL⁻¹".

6. Should the wavelength be 485 nm instead of 456 nm? This appears to be a recurring mistake throughout the manuscript (e.g., "Similarly, parallel experiments using E-formulations combined with two 456 nm/365 nm..."). Please verify and correct all instances.

Answer: Thank you for raising this point. We confirm that the wavelength reported as 456 nm in several places was indeed a typographical error. In the cell culture irradiation experiments used to promote *Z*→*E* photoswitching, the actual wavelength applied was 466 nm, not 456 nm. All corresponding instances have now been corrected in the revised manuscript. For logistical constraints: the optical filter used for irradiating cell culture plates was not the same as that employed for photochemical characterization or nanocomplex formulation (485 nm). The resulting wavelength difference was not considered limiting, as the primary aim of these experiments was to test the hypothesis that improvements in transfection efficiency might be driven by light-induced isomerization, specifically through local, dynamic nanomechanical strain within the nanocomplexes. This hypothesis was explored rather than attempting to precisely match the photostationary states observed in spectroscopic conditions.

7. Please correct the English in the Supporting Information, as I found several errors (e.g., "acquisition," "an extemporaneous solution," "the PSS is validated," etc.).

We apologize for the errors in the Supplementary Information. It has now been carefully revised to improve clarity and overall language quality in the revised version.